# Impact of Heavy Metals on Cold Acclimation of *Salix viminalis* Roots

**DOI:** 10.3390/ijms25031545

**Published:** 2024-01-26

**Authors:** Valentin Ambroise, Sylvain Legay, Marijke Jozefczak, Céline C. Leclercq, Sebastien Planchon, Jean-Francois Hausman, Jenny Renaut, Ann Cuypers, Kjell Sergeant

**Affiliations:** 1Greentech Innovation Centre (GTIC), Environmental Research and Innovation (ERIN) Department, Luxembourg Institute of Science and Technology (LIST), 5 Avenue des Hauts-Fourneaux, L-4362 Esch-sur-Alzette, Luxembourg; valentin.ambroise@outlook.com (V.A.); sylvain.legay@list.lu (S.L.); celine.leclercq@list.lu (C.C.L.); sebastien.planchon@list.lu (S.P.); jean-francois.hausman@list.lu (J.-F.H.); jenny.renaut@list.lu (J.R.); 2Centre for Environmental Sciences, Hasselt University, Agoralaan Building D, B-3590 Diepenbeek, Belgium; marijke.jozefczak@uhasselt.be (M.J.); ann.cuypers@uhasselt.be (A.C.)

**Keywords:** abiotic stress, heavy metals, frost, transcriptomics, proteomics, antioxidant system, integrative biology

## Abstract

In nature, plants are exposed to a range of climatic conditions. Those negatively impacting plant growth and survival are called abiotic stresses. Although abiotic stresses have been extensively studied separately, little is known about their interactions. Here, we investigate the impact of long-term mild metal exposure on the cold acclimation of *Salix viminalis* roots using physiological, transcriptomic, and proteomic approaches. We found that, while metal exposure significantly affected plant morphology and physiology, it did not impede cold acclimation. Cold acclimation alone increased glutathione content and glutathione reductase activity. It also resulted in the increase in transcripts and proteins belonging to the heat-shock proteins and related to the energy metabolism. Exposure to metals decreased antioxidant capacity but increased catalase and superoxide dismutase activity. It also resulted in the overexpression of transcripts and proteins related to metal homeostasis, protein folding, and the antioxidant machinery. The simultaneous exposure to both stressors resulted in effects that were not the simple addition of the effects of both stressors taken separately. At the antioxidant level, the response to both stressors was like the response to metals alone. While this should have led to a reduction of frost tolerance, this was not observed. The impact of the simultaneous exposure to metals and cold acclimation on the transcriptome was unique, while at the proteomic level the cold acclimation component seemed to be dominant. Some genes and proteins displayed positive interaction patterns. These genes and proteins were related to the mitigation and reparation of oxidative damage, sugar catabolism, and the production of lignans, trehalose, and raffinose. Interestingly, none of these genes and proteins belonged to the traditional ROS homeostasis system. These results highlight the importance of the under-studied role of lignans and the ROS damage repair and removal system in plants simultaneously exposed to multiple stressors.

## 1. Introduction

As sessile organisms, plants are exposed to changing environmental conditions. When these environmental conditions negatively impact plant growth and primary production, they are called abiotic stresses [1]. 

To cope with adverse conditions, plants have evolved a wide network of molecular mechanisms that allow them to respond to stress by adjusting their metabolism [2]. These metabolic changes are made possible by fine-tuning gene transcription and protein abundance and activity. During the last few decades, substantial progress has been made in the identification of stress-related genes and proteins and uncyphering the mechanisms involved in stress responses. However, a large majority of these studies have been carried out in laboratories under controlled conditions. Under natural conditions, plants are seldomly exposed to one single stress at a time [3]. An increasing amount of evidence from field and molecular studies have shown that plants respond to combinations of stresses in a non-additive manner, resulting in interacting effects that cannot be predicted from studies focusing on single stress exposure [3,4]. 

Another issue is the temporal scale and the degree of exposure used in stress-related studies. For example, heavy metal stress-related studies often focus on short-term responses with metal concentrations exceeding ecologically relevant concentrations [5]. However, it has been observed that plants exposed to long-term heavy metal stress present a different metabolic profile than plants exposed to acute stress [6]. 

Despite the specificity of stress responses at a gene and protein level, some common processes have been observed in the response to different stresses, including the production of reactive oxygen species (ROS) and the accumulation of hormones such as abscisic acid (ABA) and jasmonic acid (JA). Among those, [4] identified the antioxidant defence machinery as an underlying pathway leading to tolerance to stress combinations. Unravelling the crosstalk existing between the responses to different stresses could pave the way to the breeding of multiple-stress-resistant plants. 

A few years ago, our team observed that *Salix viminalis* grown in pots in a soil polluted by a mixture of heavy metals were more winter-tolerant than plants grown in unpolluted soil (unpublished results). *S. viminalis*, known as basket willow, is a fast-growing pioneering shrub with a wide Eurasian distribution. When fully cold-acclimated, its twigs can withstand up to −196 °C [7]. Given the extreme frost tolerance of its aboveground organs, it was hypothesized that the difference in winter-tolerance was the consequence of an increased frost-tolerance at the root level.

In this present study, we investigated how exposure to a mild polymetallic stress was able to increase cold acclimation and frost hardiness in the roots of *S. viminalis*. Our hypothesis was that the exposure of the roots to a low concentration of heavy metals would prime its antioxidant system, resulting in higher antioxidative capacity, itself leading to a higher root frost hardiness, in a way analogous to how cold priming alleviates acute metallic stress [8]. In order to test this hypothesis, we characterised the long-term effects of exposure to cold and a mild polymetallic contamination, combined and separate, at the transcriptome and proteome level. We also measured the enzymatic activity and the concentration of several compounds involved in redox homeostasis in the roots of *S. viminalis*. The aim of the present study was to compare the effects of a mild polymetallic mixture and/or cold acclimation on the antioxidant defence system, with an emphasis on systems displaying interaction patterns.

## 2. Results

### 2.1. Plant Material

Plants were exposed to cold and/or a polymetallic mixture, resulting in four temperature–polymetallic contamination combinations possible: control soil temperate (CT), control soil cold acclimated (CC), metal-polluted soil temperate (MT), and metal-polluted soil cold acclimated (MC).

Plants exposed to the polymetallic mixture were smaller and showed a sickly habit (Figure 1). Their leaves were chlorotic, and some necrotic spots could be observed. In addition, some of the older leaves dried on the plants. Upon cold exposure, plant growth slowed down in plants cultivated in contaminated and uncontaminated soil. No further growth was observed after three weeks of cold acclimation.

#### 2.1.1. Chlorophyll Fluorescence

The chlorophyll fluorescence of young leaves was measured at several time points during the four weeks of cold exposure, and the F_v_/F_m_ ratio was calculated. The F_v_/F_m_ ratio of plants in the CT group stayed relatively stable, with an average value of 0.82 (Appendix A). The F_v_/F_m_ ratio of MT plants was lower, with an average value of 0.71, but it fluctuated more, as shown by its standard deviation of 0.15.

Upon cold exposition, F_v_/F_m_ values temporarily decreased in plants cultivated in control and polluted soil. The average F_v_/F_m_ ratio of plants exposed to cold was 0.80 and 0.65 for plants cultivated in control and polluted soil, respectively. 

The effects of cold and polymetallic exposure on the F_v_/F_m_ ratio were assessed with a two-way ANOVA. The polymetallic exposure had a significant (*p*-value < 0.001) impact on the F_v_/F_m_ ratio, but no effect of cold exposure (*p*-value = 0.340), sampling date (*p*-value = 0.456), or interaction between cold and polymetallic exposure (*p*-value = 0.590) could be detected. 

#### 2.1.2. Root Frost Tolerance

Cold acclimation significantly decreased root electrolyte leakage (REL) at −2 °C (two-way ANOVA, *p*-value < 0.001). Indeed, while REL was over 75% in non-cold acclimated roots exposed to −2 °C, indicating that the roots were not frost-tolerant, it remained below 40% for cold-acclimated roots (Appendix A).

On the other hand, exposure to heavy metals (HM) did not significantly impact REL at −2 °C (two-way ANOVA, *p*-value = 0.228). However, it must be noted that exposure to HM reduced root growth and that material from only one plant could be harvested for plants at the control temperature, while only two biological replicates could be measured for HM plants at low temperature. Although values for the latter measurements indicate that combined exposure to cold and HM further decreased REL compared to cold alone, no significant interaction could be detected (two-way ANOVA, *p*-value = 0.728). This needs to be reassessed.

The effects of cold and polymetallic exposure were assessed with a two-way ANOVA. The polymetallic exposure had a significant (*p*-value < 0.001) impact on the F_v_/F_m_ ratio, but no effect of cold exposure (*p*-value = 0.340), sampling date (*p*-value = 0.456), or interaction between cold and polymetallic exposure (*p*-value = 0.590) could be detected.

### 2.2. Impact on the Antioxidant System

#### 2.2.1. Antioxidant Capacity

Antioxidant capacity is divided into two components: the antioxidant capacity of the hydrophilic fraction (containing glutathione, ascorbic acid, and flavonoids compounds) and the antioxidant capacity of the lipophilic fraction (containing tocopherol, zeaxanthin, and carotene, amongst others [9]). Of the three tested conditions, none significantly affect the lipophilic antioxidant capacity (Figure 2, Table 1(1)). Nevertheless, cold-acclimated samples tend to have a higher lipophilic antioxidant capacity, while metal exposure roots have a lower lipophilic antioxidant capacity. In the hydrophilic fraction, metal exposure has a significant negative impact on antioxidant capacity (Figure 2, Table 1(1)). On the other hand, cold acclimation has no significant effect on the hydrophilic antioxidant capacity. 

When the antioxidant capacity of both fractions was summed up, only metal exposure has a significant and negative impact on the total antioxidant capacity (TAC) of the roots (Figure 2, Table 1(1)). While cold increased the average TAC of the root samples, the high variability observed makes this trend non-significant.

#### 2.2.2. ROS Scavenging Enzymes Activity

The activity of three ROS scavenging enzymes was studied. (1) Catalase (CAT) activity is significantly and negatively affected by cold acclimation, while it increases significantly after exposure to heavy metals (Figure 2, Table 1(2)). The exposure to both conditions simultaneously does not have any significant interaction effect on the CAT activity. (2) The activity of glutathione reductase (GR) is not significantly affected by exposure to a single stress, but simultaneous exposure has a significant interaction effect (Figure 2, Table 1(2)). (3) Superoxide dismutase (SOD) activity is significantly and negatively affected by metals (Figure 2, Table 1(2)), but not by cold acclimation. 

#### 2.2.3. Glutathione

As shown in Figure 2, cold acclimation has a strong and significant effect on the total glutathione content. While metals tend to increase the total glutathione concentration, its effect is non-significant. 

Similar results can be observed for glutathione (GSH) and oxidized glutathione (GSSG) (Figure 2, Table 1(3)). Cold acclimation significantly increases their concentration. However, while metals also tend to increase GSSG content, they have the opposite effect on GSH. Interestingly, no significant result could be observed for the ratio between oxidised and total glutathione content (Figure 2, Table 1(3)), but metals tend to decrease the percentage of reduced GSH.

### 2.3. Transcriptomics

#### 2.3.1. De Novo Transcriptome Assembly, Functional Annotation, and Mapping

The de novo transcriptome of *S. viminalis* was obtained by merging and assembling the reads obtained by sequencing cDNA libraries of fine roots exposed to cold and/or a polymetallic mixture.

A total of 607,614,886 of 150-base pair-sequence reads were obtained, with libraries ranging from 40 to 57 million reads. After trimming and filtering, approximately 480 million reads remained. The merging and filtering of the de novo assembled transcriptomes generated 86,843 unique contigs. Of those, approximately 75% (64,173) were successfully annotated using *S. purpurea* as reference. More than 75% of the filtered reads mapped back to the de novo assembled transcriptome, indicating a good quality assembly [10]. Between 17% and 20% of the reads mapped to multiple contigs. This large proportion of reads mapped to multiple contigs likely arises from the whole-genome duplication event, known as the “salicoid” duplication, which occurred 58 Mya [11].

#### 2.3.2. Differentially Expressed Genes

In order to determine differentially expressed genes (DEGs), the R-based package edgeR (v. 3.28.1 [12]) was used. As recommended, low-expressed contigs were filtered out because they have a low probability of playing a biological role and by necessity due to statistical approximations used by the edgeR algorithm. The filtering threshold was fixed at 0.5 counts per million (representing on average 20 counts per library) in at least three libraries. After filtering, 43,169 contigs remained. 

Multidimensional scaling analysis of the data resulted in a good separation of the treatments (Figure 3a). Cold acclimation and exposure to metals appeared to have a distinct effect on gene expression. Based on this analysis, the effect of exposure to both stresses generated a distinct expression profile compared to the other conditions. 

Of the 43,169 contigs analysed, 6722 contigs were differentially expressed (FDR-adjusted *p*-value < 0.01) compared to the control in at least one of the conditions (Figure 3b). In CC roots, 1059 and 457 transcripts were down- and upregulated; in MT roots, 3016 and 1321 transcripts were down- and upregulated; and in MC roots, 1233 and 1194 transcripts were down- and upregulated. BLAST analysis of these 6722 unique contigs against the TAIR database resulted in 3960 annotations. For these, a TAIR name was obtained. 

Only 67 contigs were significantly regulated at the intersection of the three conditions (Figure 3b). From the three conditions, exposure to heavy metals impacted the expression of the most genes (4337 genes up- or downregulated), followed by the simultaneous exposure to the polymetallic mixture and cold acclimation (2427) and cold acclimation (1516). It seems that cold acclimation reduced the impact of exposure to the polymetallic mixture on gene expression. 

#### 2.3.3. Gene Ontology Analysis

In order to identify the ontologies most involved in each condition, a gene ontology (GO) enrichment analysis was performed using clusterProfiler (v. 3.14.3 [13]). All three conditions of interest showed significant GO term enrichment (*p* < 0.05) for genes both down- and upregulated (Figure 4).

Using the 3960 contigs for which a GO annotation was obtained, the GO enrichment analysis revealed that processes related to the response to sucrose, catabolism of organic aromatic and cyclic compounds, and response to water deprivation were downregulated in plants exposed to cold (Figure 4). Conversely, processes related to polysaccharide catabolism were upregulated, along with mRNA metabolic processes. 

In the roots of willows exposed to metals, processes related to the catabolism of organic aromatic and cyclic compounds as well as responses to sucrose and processes related to cell cycle regulation and gene silencing were downregulated. Upregulated GO terms were associated with response to water deprivation, cell death, metal ion homeostasis, and detoxification (Figure 4).

In plants exposed to both stresses, the GO terms that were significantly downregulated corresponded to those downregulated by cold acclimation alone, with the addition of the cell death GO term. Contrastingly, upregulated processes were similar to those upregulated in willows exposed to single conditions. These processes were related to metal ion homeostasis, carbohydrate catabolism, plant-type hypersensitive response, and cell death (Figure 4). Interestingly, when filtering GO terms to levels 6 and 7, it appeared that root hair cell development was downregulated under all three conditions. 

### 2.4. Proteomics 

A total of 895 proteins (884 with TAIR annotation) were identified using our in-house database and the criteria described in the Section 4. Of those, 390 proteins (387 with TAIR annotation) were found to be significantly differentially abundant (absolute log FC ≥ 1.5 and FDR adjusted *p*-value ≤ 0.05). 

In order to unravel the effect of cold acclimation and long-term polymetallic exposure on the root proteome, these proteins were clustered. Protein abundance was normalised by the average control abundance, then Eisen’s cosine as a distance metric was used, followed by hierarchical clustering with complete linkage. Using a correlation threshold of 0.75, seven clusters were obtained (Figure 5a).

Cluster 1 is composed of proteins that are downregulated in all conditions. Although it is composed of only three proteins (NRT2.1, GAPDH C2, and a calcineurin-like metallo-phosphoesterase protein), clusterProfiler determined that GO terms associated with response to nitrate, gluconeogenesis, and cadmium (Cd) ions were significantly enriched in this cluster (Figure 5b, Appendix A).

Cluster 2 comprises proteins that are more abundant in MT roots but relatively stable in CC and MC roots. GO terms associated with response to insects, and nucleobase-containing small molecule metabolic processes, are enriched in this cluster (Appendix A). 

Opposite to cluster 2, cluster 3 is characterised by a lower protein abundance under MT and an abundance close to the control level under CC and MC. Its GO terms can be divided into three distinct groups: those related to brassinosteroid biosynthetic processes, those related to negative translation regulation, and those related to the proteasome catabolic process (Appendix A).

The abundance of proteins in cluster 4 was significantly increased under CC and MC, while it decreased under MT. As shown in Figure 5b, GO terms related to response to Cd, ribosome assembly, and metal ion homeostasis are significantly enriched in this cluster. In addition, GO terms related to aromatic amino acid metabolism are also enriched (Appendix A).

Cluster 5 presents a similar abundance profile as cluster 4. It is characterised by GO terms associated with response to Cd and ribosome assembly, but also to acetyl-CoA, nucleotides, flavonoid and phenylpropanoid metabolism, translation elongation, and proton transmembrane transport (Figure 5b and Appendix A).

The abundance of proteins in cluster 6 remains relatively close to the control level. In this cluster, GO terms related to protein refolding and amino acid metabolism are significantly enriched. In addition, as shown in Appendix A, GO terms related to response to Cd, fatty acid metabolism, response to fungus, and glucose metabolism were also enriched.

The proteins in cluster 7 have a positive interaction: while CC and MT roots have protein abundance close to the control level, protein abundance in MC roots significantly increases. GO terms enriched in this cluster can be divided into five groups: those related to cold acclimation, response to Cd, response to ROS, phenylpropanoid metabolism, and JA metabolism (Appendix A). 

## 3. Discussion

*Salix viminalis* can withstand various abiotic stresses, amongst which are exposure to high concentrations of metals and freezing. In this study, cuttings of three-month-old *S. viminalis* clones were exposed to either cold, a polymetallic mixture, or the combination of both in a controlled environment to reduce experimental variation. Aboveground organs were monitored during the cold exposure to evaluate the progression of the cold acclimation process.

### 3.1. Impact of Cold Acclimation

Cold exposure was sufficient to reduce and even stop plant growth, as previously observed in poplar [14], and *Juniperus chinensis* [15], amongst others. Although the F_v_/F_m_ reduction was significant, it was smaller than in poplar exposed to 4 °C [14]. The roots were able to cold acclimate, as shown by the decrease in REL at −2 °C (Appendix A). However, the gain in frost hardiness was lower than what has been reported previously [16]. This can originate from two factors. First, the plants used in this study were young cuttings that had almost exclusively fine white roots. Younger plants tend to cold acclimate to a lesser degree than older ones, and fine roots are more frost sensitive than coarse roots [17]. Second, to focus on the effect of cold, the photoperiod was kept at 16 h/8 h, while under natural conditions the temperature decrease leading to cold acclimation coincides with a reduction of the photoperiod. Although the photoperiod does not directly impact roots, it could play a role in root cold acclimation by changing the source-sink relationship between roots and aboveground organs [18]. 

Cold acclimation increased the abundance of various proteins (in clusters 4, 5, 6, and 7 in Figure 5a). These include heat-shock proteins (HSP) 70 and 90, molecular chaperones during cold stress; dehydrins and proteins related to polyamines synthesis (S-adenosylmethionine synthetase), known to stabilise membranes and proteins during cold stress; and peroxidase involved in ROS detoxification. Proteins related to the energy metabolism, such as sucrose synthase, GAPDH, ATPase, and alcohol dehydrogenase, increased in abundance. These proteins are more abundant in cold-exposed chicory roots [19] and poplar [14], amongst others. 

Few studies focus on long-term cold-induced transcriptomic changes. However, GO enrichment analysis is consistent with what is observed in previous studies. Starch and polysaccharide catabolism GO terms were expected as cold acclimation is associated with the conversion of starch into simple sugars [14,15,20]. Cold acclimation is also associated with a transient increase in the basal metabolism followed by a reduction of it. GO terms related to “response to water” were expected, as an essential function of cold acclimation is to mitigate upcoming frost-induced osmotic stress [16,21]. Although they play a predominant role in cold acclimation [22,23], no CBFs (genes commonly linked to cold signalling) were found in the DEGs. This might be because CBFs are highly induced upon cold perception but gradually return to control levels as exposure time increases. Kreps et al. [24] observed that *CBF1*, *2*, and *3* were induced (respectively 2.5-fold, 22-fold, and 30-fold), in roots of *Arabidopsis thaliana* 3 h after cold exposure, but only 0.73-fold, 2.5-fold, and 7-fold after 27 h. 

Cold acclimation did not significantly increase CAT and SOD activity. This is contrary to observations in the roots of cold acclimated *Cichorium intybus* [19], *Triticum aestivum* [25], and Cicer arietinum [26]. Nevertheless, GR activity and GSH levels significantly increased in cold acclimated roots (Figure 2), as observed in *Pinus banksiana* [27]. 

### 3.2. Impact of Long-Term Metal Exposure

Exposure to the polymetallic mixture strongly impacted plant growth and habit (Figure 1). Plants exposed to metals were shorter and chlorotic, as previously observed in multiple species [28,29,30]. The exposure to the polymetallic mixture also significantly lowered F_v_/F_m_, consistent with what has been observed previously in the same clone [31] and other *Salix* species [32]. 

Several protein families were more abundant in HM-exposed roots. These proteins, grouped in clusters 2, 6, and 7 (Figure 5a), were involved in protein synthesis and folding (HSPs 70 and 90 and ribosomal proteins), membrane transporters, SAM-related proteins, and Kunitz trypsin inhibitors. All these proteins were already found to be more abundant in poplar exposed to Cd for 56 days [28] and in various *S. fragilis* × *alba* clones exposed to metal-contaminated sediments [32]. Sulphite reductase, involved in sulphate assimilation, was also more abundant in MT roots. Sulphur compounds play an important role in mitigating various stresses, including metal stress, via their action on ROS detoxification and metal chelation and sequestration [33]. Interestingly, some transcripts related to sulphate assimilation were more abundant in MC roots than in MT roots and followed a positive interaction pattern. Various sulphur assimilation-related proteins have been reported to be more abundant in the roots of *A. thaliana* [34] and *Triticum aestivum* [35] under Cd stress. Peroxidase, CAT, and glutathione-S-transferases were also more abundant in MT roots, as previously reported in poplar [28]. Chromatin remodelling proteins (histones, TF jumonji family protein, pds5-related proteins) were also more abundant in MT roots. The accumulation of these proteins is consistent with the impact of metals on the expression of the genes involved in the cell cycle and histone modification, as observed in the transcriptomics analysis (Figure 4). Metals, and in particular Cd, are known to impact the cell cycle. Monteiro et al. observed that the root cells of *Lactuca sativa* exposed to 1 µM Cd tended to be blocked at the G2 checkpoint [36], and Hendrix et al. similarly observed that Cd inhibits cell division and endoreduplication in the leaves of *A. thaliana* exposed to 5 µM Cd [37]. Several proteins involved in cell wall biosynthesis (cellulose synthase 1, 4, and 6, GPAT8) were also more abundant. The cell wall can play a barrier role against Cd and reduce Cd absorption and translocation, via Cd chelation, in several plant phyla [38,39].

GO terms associated with metal ion homeostasis, detoxication, cell death, and water deprivation were enriched in MT roots. The first three terms are the direct consequence of the toxicity of the polymetallic mixture and the different mechanisms used by the plant to reduce the deleterious effects of exposure [33,40]. On the other hand, water deprivation is often observed in metal-exposed plants and could be the result of the impact of metals on root growth and aquaporins [39].

Exposure to the polymetallic mixture significantly increased CAT and GR activity and decreased SOD activity. A decrease in SOD activity was observed in Cd-exposed poplar [28], where it was hypothesised to be due to a lower Cu and Zn uptake induced by Cd exposure. However, in this study, Cu and Zn were also supplemented in the soil, indicating that the reduction in SOD activity is not linked to a decreased availability of Cu and Zn. Smeets et al. observed no significant variation in SOD activity in the roots of *A. thaliana* exposed to Cd and/or Cu [41]. In addition, they observed that CAT activity was not significantly influenced by the simultaneous exposure to Cd and Cu but that such exposure significantly reduced GR activity, contrary to what was observed here. Conversely, Tauqeer et al. [42] observed an increase in APX, CAT, POD, and SOD capacity in the roots of *Alternanthera bettzickiana* exposed to Cd for eight weeks, although this tended to be lower after exposure to 2 mM Cd than after exposure to 1 mM Cd. Similar results were observed in four cultivars of alfalfa exposed to Zn [43]. This indicates that plants have different strategies to cope with long-term metal stress. The exposure to metals also significantly reduced TAC, as observed in *Solanum tuberosum* and *Allium cepa* grown in metal-contaminated soil [44]. This decrease in TAC was not linked to reduced GSH levels as there is no difference in reduced GSH concentration between CT and MT roots.

### 3.3. Simultaneous Exposure to Cold and Metals

The simultaneous exposure to cold and metals did not further decrease plant growth, nor was there a further increase in chlorosis. In addition, no interaction effects could be observed in F_v_/F_m_. Likewise, no increase in REL at 4 °C was observed, although metals are known to increase electrolyte leakage via lipid peroxidation [42,45,46]. No significant difference in REL at −2 °C between CC and MC could be observed. This contrasts with what Talanova et al. [47] reported in Cd-exposed wheat. They observed that exposure to Cd and cold significantly increases the frost hardiness of wheat leaves after one day, but frost hardiness returned to control levels after three days.

Contrarily to what was hypothesized, the increased frost tolerance that was previously observed in *S. viminalis* roots exposed to cold and the polymetallic mixture could not be explained by a priming effect of heavy metals on the ROS detoxification capacity of the roots. Indeed, based on the measures of the antioxidant system, it seems that the polymetallic mixture impacted more strongly and more negatively the antioxidant system than cold acclimation did (Figure 2). In addition, CAT, GR, GSH1, and GSH2 were not induced at a transcriptomic or proteomic level, and while SOD transcripts were upregulated in MT and MC roots, it did not result in an increase in SOD activity. Other processes that could explain the increased frost hardiness were investigated.

Frost hardiness is a complex trait that can arise from multiple metabolic adaptations. The most commonly proposed metabolic changes leading to increased frost hardiness are (1) the increase in the concentration of small saccharides (glucose, fructose, and trehalose), (2) modification of the cell wall, (3) an increase in the unsaturation of the membrane lipids, (4) an increase in the activity of the ROS homeostasis machinery, and (5) the production of protective molecules such as polyamines and dehydrins [21]. The potential impact of these mechanisms on cold acclimation in the roots of plants exposed to a polymetallic mixture will be assessed based on our transcriptomic, proteomic, and enzymatic activity results.

With the increase in the ROS homeostasis machinery being ruled out as a possible explanation, we will now focus on the four remaining candidate processes related to frost hardiness. Given the vast amount of data generated by this study, a special emphasis will be placed on the interactions observed between cold acclimation and metal exposure. Indeed, the impact of metals on roots being stronger on the antioxidant capacity than the impact of cold acclimation, we wanted to find processes that could explain the recovery of frost hardiness despite the strong negative impact of metals on ROS homeostasis. Several nomenclatures to describe interaction patterns have been established; for the sake of clarity, the one described by Rasmusen et al. [48] will be used.

When looking at the transcriptome, different gene response patterns can be distinguished. These patterns are as follows: combinatorial (genes respond similarly for single conditions but not when exposed to both), cancelled (genes respond to at least one condition, but there is no response when exposed to both conditions), prioritised (both single conditions have an opposite effect and one response is prioritised when exposed to both conditions), independent (one single and the simultaneous exposure lead to the same response, the other condition does not influence gene response), and similar (all responses are similar). 

Similarly to what Rasmusen et al. [48] observed in *Arabidopsis* exposed to multiple stresses, the cancelled, independent, and combinatorial responses were the most common, representing 63.1%, 19.2%, and 16.3% of the DEGs, respectively. The cancelled pattern corresponds to an antagonistic response at the expression level. Similarly, the prioritised (0.4%) pattern reflects an antagonist response where the response to one stress is prioritised over the other. Conversely, combinatorial and similar (1%) patterns represent positive interactions in the responses to different stresses. Finally, the independent pattern is observed in genes whose activation is entirely due to one stress and not the other. The difference between the prevalence of the cancelled response in this study and what was observed in *Arabidopsis* (30.6%) likely comes from a methodological difference. While Rasmusen et al. [48] used the top 500 DEGs for the 11 conditions studied, we used all the DEGs, even those with a low abundance. Once this difference is considered, our results are in line with the results of Rasmusen et al. [48]. 

While a large proportion of the DEGs is specific to simultaneous exposure to both metals and cold, no new biological process GO term arose in the GO analysis of the transcriptome. Therefore, the combinatorial pattern was not observed at the GO level, suggesting a high level of functional overlap at the gene level. Most GO terms had an independent or similar pattern (present 9 and 4 times in the 30 most significant GO terms, respectively). This is because most GO terms were enriched in only one of the single exposure conditions.

The prioritised pattern was the least represented at the transcriptome level; however, three GO terms displayed this pattern. Nine GO terms presented the cancelled expression pattern. Interestingly, out of those, seven were downregulated by metals and upregulated in the interaction group. Together, this indicates that while it is generally impossible to predict the interaction pattern of a single transcript, at the functional level, based on “biological process” GO terms, the predictability increases. Finally, four GO terms could not be classified as they were found to be simultaneously up- and downregulated.

The relationship between correlated and anti-correlated genes and proteins exhibits a similar pattern. While only five TAIR annotations were observed in both positively and negatively correlated genes and proteins, most of the GO terms found in the negatively correlated group were also found in the positively correlated group. Two exceptions to this observation are the GO terms associated with protein folding and chemical homeostasis. 

When applying the same classification pattern to the clustered proteomic data, the following results were obtained: cluster 1 is similar, cluster 2 is independent, cluster 3 is cancelled, clusters 4 and 5 are prioritised, and cluster 7 is combinatorial. Proteins in cluster 6 remain close to control values in all conditions but show significant differences between the treatments; these do not show an interaction pattern. Generally, the proteome profile of MC roots was closer to that of CC roots than that of MT roots. Indeed, both conditions had similar protein abundance in clusters 1, 3, 4, 5, and 6, representing a total of 286 out of 335 proteins (Figure 5a). This might partially explain why MC roots have REL values similar to CC roots at −2 °C. 

Several proteins were most abundant in roots simultaneously exposed to both stresses. These proteins were found in clusters 4, 5, 6, and 7 (Figure 5a). Of those clusters, the last is of particular interest as it shows a strong synergetic interaction pattern, indicating that simultaneous exposure increased the abundance of these proteins more than expected from a mere additive effect. These proteins are of interest as their overexpression could lead to plants more resistant to frost and heavy metals simultaneously. When looking at the precise GO terms associated with this cluster (Appendix A), five major groups can be highlighted: cold acclimation (associated with response to water and carbohydrate catabolic process), response to Cd, response to ROS, phenylpropanoid biosynthetic process, and JA biosynthetic process. 

The first two GO terms, cold acclimation and response to Cd, are responses to cold and HM treatment, respectively. Since these proteins show a strong synergetic effect, they could be important in the resistance to multiple stresses. The proteins associated with the cold acclimation GO term were AT1G56070 (LOS, translation elongation factor 2-like protein), AT1G20450 (ERD10, dehydrin), AT1G10760 (SEX1, α-glucan water dikinase), and AT3G02360 (6PGD2, 6-phosphogluconate dehydrogenase). The proteins associated with the response to the Cd ion GO term were AT1G56450 (PBG1, beta subunit G1 of the 20S proteasome), AT3G48000 (ALDH2, aldehyde dehydrogenase), AT1G60710 (ATB2, aldo-keto reductase), AT1G07920 (a protein belonging to the elongation factor-tu family), AT1G77120 (ADH1, alcohol dehydrogenase), and AT4G39230 (PCBER1, a phenylcoumaran benzylic ether reductase).

The three remaining GO terms were associated with a more general abiotic stress response. The first of those was the response to oxidative stress GO term. Initial disturbance in the ROS homeostasis, followed by an increase in the ROS scavenging capacity, has been observed in response to most abiotic stresses [49,50]. The five proteins associated with this GO term were AT5G67400 (RHS19, a protein with expected peroxidase activity), AT5G58400 (a peroxidase), AT3G13160 (RPPR3B, a ribosomal pentatricopeptide repeat protein), AT1G77120 (ADH1, also related to cold acclimation), and AT2G06050 (OPR3, a 12-oxophytodienoate reductase, also involved in JA biosynthesis). 

Similarly, the biosynthesis of JA has been observed as a response to multiple abiotic stresses. Furthermore, the external application of JA increases plant resistance to chilling, freezing, and heavy-metal stress, as reviewed [51]. The proteins associated with this GO term were AT2G06050 (OPR3, also found in response to oxidative stress) and AT1G55020 (LOX1, a lipoxygenase catalysing the first step of JA biosynthesis). 

The least-significant GO term, phenylpropanoid biosynthesis, has also been observed in response to several stresses, including cold [52] and heavy metals [53]. The proteins linked to this GO term were AT3G13610 (F6’H1, an oxoglutarate-dependent dioxygenase), AT4G39230 (PCBER1, also found in response to Cd), AT3G53480 (ABC G3, an auxin transporter involved in the export of phenolic compounds), and AT1G64160 (DIR5, a protein involved in the synthesis of pinoresinol). Gutsch et al. observed that the cell wall proteome of *M. sativa* exposed to long-term Cd exposure is enriched in proteins related to the phenylpropanoid pathway and lignification [54]. However, they also observed that while the phenylpropanoid pathway was induced in alfalfa stems exposed to long-term Cd, this resulted in the accumulation of secondary metabolites, such as lignan and flavonoids, rather than in increased lignification [54]. In this current study, the synthesis of lignans seemed to be favoured, as shown by the overabundance of PCBER1 and DIR5. PCBER1 is known to be abundant in the xylem of poplar [55] and DIR proteins could play a role in the lignification of casparian strips [56]. In addition, based on our transcriptomics data, the genes involved in the first steps of the phenylpropanoid pathway were not upregulated in MC roots. However, the first two genes involved in the flavonoid biosynthetic pathway (*CHS* and *CHI*) were downregulated while the genes leading to the formations of lignans (including *HCT*, *CAD*, *PCBER1*, and *DIR5*) were upregulated. Altogether, this indicates a shift from the biosynthesis of flavonoids toward the production of lignans.

Of the five tolerance mechanisms mentioned above, four have been encountered in the presented data so far: the adaptation of the ROS homeostasis machinery, the modification of the cell wall (potentially via PCBER1 and DIR5), the production of protective molecules including dehydrins (ERD10), and small saccharides (through SEX1 and 6PGD2). Interestingly, transcript coding for raffinose synthesis (*DIN10* and *SIP1*), an oligosaccharide whose accumulation is related to root frost hardiness in alfalfa [57], were found to be more abundant in MC roots, while they were downregulated in the roots exposed to the other treatments, although the difference was non-significant. A similar pattern was observed in the transcripts coding for trehalose-6-phosphate synthase (*TPS*) and trehalose-6-phosphate phosphatase (*TPP*). These are involved in the production of trehalose, a disaccharide with strong frost-protecting effects [21].

In addition to these mechanisms, more general stress responses were observed in the proteome data, namely the accumulation of the 20S proteasome subunit, HSP 90, and translation elongation factors. 20S proteasome is involved in ubiquitin-independent protein degradation and plays an important role during oxidative stress [58]. HSPs are molecular chaperones that prevent protein misfolding. They are known to be important in multiple stresses, including cold and heavy metals [59]. Finally, translation elongation factors function in protein synthesis, but they also have chaperone activity and participate in the 26S proteasome [60,61]. All these proteins support the importance of the protein repair and removal mechanisms during multiple stress exposure, leading to increased ROS burden, as highlighted by Mittler [49]. Furthermore, transcripts related to sulphate assimilation were also found to be more abundant in MC roots and followed a positive interaction pattern. As mentioned above, sulphur compounds are known to play an important role in several stress-tolerance mechanisms [33].

## 4. Materials and Methods

### 4.1. Plant Materials

Cuttings of 10 cm from one individual *S. viminalis* were rooted in containers filled with a mix of 25 kg of potting soil and 17.5 kg sand under a controlled environment (23/20 °C, 60% relative humidity, 16/8 h photoperiod). After three weeks, plantlets were transferred to individual pots. Half of those pots contained soil that had been spiked with a mixture of heavy metals provided in the form of metal chlorides (final concentrations of 1.5 mg/kg of dry soil Cd, 175 mg/kg of dry soil Cu, 30 mg/kg of dry soil Ni, 500 mg/kg of dry soil Zn). After 40 days, half of the plants were randomly chosen and cold-acclimated (7/5 °C, 60% relative humidity, 16/8 h photoperiod), while the remaining plants were kept under control conditions. A total of four temperature–polymetallic contamination combinations were obtained, namely control soil temperate, control soil cold acclimated, metal-polluted soil temperate, and metal-polluted soil cold acclimated. Three biological replicates per stress combination were analysed. During the four weeks of cold treatment, chlorophyll fluorescence was measured at several time points, and the F_v_/F_m_ ratio was calculated. After four weeks, plants were uprooted and the roots were rinsed. Approximately one gram of roots was used to determine root frost tolerance and the remaining was snap-frozen in liquid nitrogen and stored at −80 °C until subsequent use. 

Root frost tolerance was assessed using root electrolyte leakage [62]. Briefly, for each biological replicate roots were washed using Elix-purified water, divided into five (100 to 350 mg), and placed into capped test tubes, and a drop of Elix-purified water was added to prevent desiccation and supercooling. For each biological replicate, one sample was kept at 4 °C and the remaining were exposed to 0 °C for 2 h. After that, the temperature was decreased at a rate of 5 °C h^−1^ to a minimum temperature of −10 °C. After reaching the selected temperature, the test tubes were removed and placed at 4 °C for thawing. After 2 h thawing, each tube was supplemented with 12 mL of Elix-purified water and left overnight at room temperature on a shaker. The conductivity of the bathing solution (E1) was measured using a conductivity probe (CyberScan CON 400, Eutech Instrument, Thermo Scientific, Villebon-Sur-Yvette, France). Root samples were then autoclaved at 121 °C for 10 min, and the conductivity of the bathing solution (E2) was re-measured the next day. The electrical conductivity of the Elix-purified water was also measured (E0). Root electrolyte leakage was calculated as: $$REL=\frac{E1-E0}{E2-E0}$$

### 4.2. Impact on the Antioxidant System

#### 4.2.1. Antioxidant Capacity

Analysis of the antioxidant capacity was performed using the ferric-reducing antioxidant power (FRAP) assay on hydrophilic and lipophilic fractions [63]. Briefly, 1 mL of 80% extraction buffer was added to 75 mg finely ground roots, vortexed, placed on ice, and centrifuged (18,000× *g*, 30 min, 4 °C). Antioxidant capacity was measure at 593 nm after 10 min, using TPTZ (2,4,6 Tris (2 pyridyl) s-triazine) as a substrate and trolox (6 Hydroxy-2,5,7,8 tetramethylchrommane-2-carboxylic acid) as a standard. Total antioxidant capacity was calculated as the sum of the hydrophilic and lipophilic fractions. 

#### 4.2.2. ROS Scavenging Enzymes Activity

The enzymatic activity of catalase, glutathione reductase, and superoxide dismutase were determined spectrophotometrically. Briefly, approximately 100 mg of roots were finely ground in liquid nitrogen and added to 2 mL of extraction buffer (0.1 M Tris-HCl at pH 7.8, 1 mM DTT, 1 mM EDTA). After homogenisation, samples were filtered and centrifugated (10 min, 13,500× *g*, 4 °C). Enzyme activity was measured by colorimetry at 25 °C in the supernatant. Catalase activity was measured as the rate of decomposition of hydrogen peroxide during 10 min, measured at 240 nm, GR activity was measured at 340 nm based on the reduction of GSSG using NADPH, and the activity of SOD was measured at 550 nm using the xanthine/cytochrome c method [64].

#### 4.2.3. Glutathione Redox Couples

Glutathione redox couples were measured by spectrophotometry, as described in [65]. Glutathione was extracted from finely ground roots under acidic conditions using 200 mM HCl. Extracts were centrifuged (20,000× *g*, 10 min, 4 °C) and the pH of the supernatant was brought to 4.5 using 200 mM of NaH_2_PO_4_. Each sample was separated into two fractions; one of these was incubated with 2-vinylpyridine to determine GSSG concentrations. The other was used for total glutathione concentration, determined using the kinetics of 5,5′-dithiobis-(2-nitrobenzoic acid) reduction followed at 412 nm.

### 4.3. Transcriptomics

#### 4.3.1. RNA extraction, Library Preparation, Sequencing, and De Novo Transcriptome Assembly

The precise procedure can be found in Appendix A. Briefly, root samples (250 mg) were ground into a fine powder using a mortar and pestle in liquid nitrogen. Total RNA was extracted following a modified CTAB buffer extraction protocol [66] and cleaned using the RNeasy plant mini kit (Qiagen, Leusden, The Netherlands), according to the manufacturer’s instructions, using an extra DNAse I treatment to remove genomic DNA. RNA purity and quantity were assessed using a Nanodrop ND1000 spectrophotometer (Thermo Scientific) and total RNA integrity was assessed using the RNA Nano 6000 assay (Agilent Technologies, Diegem, Belgium) and a 2100 Bioanalyzer (Agilent Technologies). All RNAs had RIN above 7.

Libraries were prepared from 5 µg of total RNA using a SMARTer Stranded RNA-Seq kit (Takara Bio Inc., Mountain View, CA, USA), following the manufacturer’s instructions. After ligation of the cDNA to an Illumina Indexing Primer Set (Takara Bio Inc.), an enrichment step was carried out using 13 cycles of PCR. The pooled libraries were sequenced on an Illumina NextSeq500 (Illumina NextSeq 500/550 High Output Kit v2.5 (300 cycles)) to generate 150 base-pair pair-end reads. Raw sequences have been deposited at the Gene Expression omnibus as accession GSE218490. The generated FASTQ files were imported into CLC Genomics Workbench v.11.0.1. Sequences were filtered and trimmed before transcriptome assembly.

De novo assembly was performed using the CLC genomics Workbench (v. 11.0.1) and Trinity [10,67], and the obtained assemblies were merged and filtered using cd-hit-est [68,69]) and TransDecoder (v. 5.5.0, https://github.com/TransDecoder/TransDecoder, accessed on 19 October 2021). The thus-selected transcripts were blasted against the annotated *S. purpurea* transcriptome (US Department of Energy Joint Genome Institute, Berkeley, California, USA), obtained from Phytozome (v. 12.1 [70], https://phytozome.jgi.doe.gov/, accessed on 19 October 2021). Top hit sequences were filtered at 10^−5^. Finally, filtered reads were mapped against the de novo assembled transcriptome using the CLC genomics Workbench. A summary of the result of these steps and their optimization can be found in Appendix A. 

#### 4.3.2. Differentially Expressed Genes and Gene Ontology Analysis

To identify DEGs, count tables generated by CLC genomics were analysed using edgeR after filtering out low-expressed transcripts (count per million < 0.5). Two statistical models were used depending on the aim of the analysis. To detect DEGs, the following model was used:Direct effect=Condition of interest−CT
where Condition of interest is either CC, MT, or MC.

Genes were considered differentially expressed when the absolute value of their adjusted fold-change was greater than 2 and their FDR-adjusted *p*-value was lower than 0.01. Gene ontology enrichment analysis was performed on DEGs using clusterProfiler to highlight significantly regulated biological processes (FDR-adjusted *p*-value < 0.05).

### 4.4. Proteomics

#### 4.4.1. Protein Extraction and Gel-Free Proteomics

Soluble protein extraction was carried out as described previously [71], with modifications. The precise procedure can be found in Appendix A. An amount of 400 mg of root were powdered in liquid nitrogen, suspended in ice-cold 10% TCA in acetone with 0.07% DTT, and centrifuged. The pellets were washed thrice with ice-cold acetone, dried, and resuspended in 1.4 mL SDS buffer/Tris-saturated phenol (pH 8.0, 1:1 *v*/*v*). After centrifugation, 5 volumes of ice-cold 0.1 M ammonium acetate in methanol were added to the upper phase. After 2 h at −20 °C, proteins were precipitated by centrifugation and washed twice with ice-cold 0.1 M ammonium acetate in methanol and twice with 80% ice-cold acetone before being dried. Dried samples were then re-solubilised in labelling buffer. Protein concentration was determined with the Bradford method [72].

Total proteins were loaded and separated for a short time of migration on 1D gels (Criterion, Bio-Rad, Temse, Belgium). Gels were stained with Instant Blue (Gentaur BVBA, Kampenhout, Belgium). Each sample was divided into two halves (bands of A—high molecular weight and B—low molecular weight), which were destained, reduced, alkylated, and digested using trypsin. The peptides were analysed using a NanoLC 425 Eksigent System (Sciex, Foster City, CA, USA) coupled to a TripleTOF 6600 mass spectrometer (Sciex). A MS survey scan from 300 to 1250 *m*/*z* with 250 ms of accumulation time was followed by 30 MS/MS scans (mass range 100–1500 *m*/*z*).

#### 4.4.2. Data Analysis

Raw data were imported into Progenesis QI for Proteomics data analysis software (v. 4, Nonlinear Dynamics, Waters, Newcastle upon Tyne, UK). Spectra were processed by Mascot (v. 2.6.0, Matrix Science, London, UK) by searching against a custom-built database using our de novo-assembled transcriptome with the following search parameters: peptide tolerance of 20 ppm; fragment mass tolerance of 0.5 Da; carbamidomethylation of cysteine as fixed modification; and oxidation of methionine, N-terminal acetylation, and tryptophan to kynurenine as variable modifications. Proteins identified with a confidence of 95% were kept for further analysis. The fractions A and B were subsequently recombined. Proteins were considered significantly different between conditions when there were at least 2 significantly identified peptides per protein, of which one is unique, for the identified protein, ANOVA *p*-value < 0.05 and a fold change > 1.5. 

The mass spectrometry proteomics data have been deposited at the ProteomeXchange Consortium via the PRIDE [73] partner repository with the dataset identifiers PXD030968 and 10.6019/PXD030968.

## 5. Conclusions

Our study investigated the interactions in the responses of *S. viminalis* roots to cold acclimation and/or exposure to a polymetallic mixture. The plants were exposed to ecologically relevant temperatures and heavy metal concentrations, at a temporal scale emphasising a long-term stress response. 

While roots exposed to heavy metals are known to have a higher degree of electrolyte leakage, simultaneous exposure to both stresses resulted in a root electrolyte leakage level comparable to cold-acclimated roots. However, unlike the hypothesis that was the basis of this work, the low root electrolyte leakage was not linked to a metal-priming effect on the ROS scavenging capacity. Indeed, the antioxidant system of roots simultaneously exposed to both stressors was highly similar to the one of roots exposed to the polymetallic mixture alone. To understand how the roots simultaneously exposed to cold and a polymetallic mixture were still able to increase their frost tolerance despite the reduced ROS scavenging capacity, we studied their responses at the transcriptome and proteome level. A special focus was put on genes and proteins displaying interactive patterns.

At the level of the individual genes, the impact of simultaneous exposure to both stresses could not be predicted based on the analysis of the gene expression of roots exposed to each stress alone, thereby confirming that the effects of exposure to multiple stresses are not the mere addition of effects of single stresses. However, from a biological function perspective, represented by biological process GO terms, a higher degree of predictability is obtained. However, this observation requires further study and should include more diverse stress treatments.

When looking at the proteome, cold acclimation appears to be the dominant determinant in roots exposed to both conditions simultaneously. A cluster of 33 proteins was of particular interest as their abundance increased in roots exposed to both conditions simultaneously but was close to the control level in roots exposed to a single condition. Roots simultaneously exposed to the two stressors also had a higher level of PCBER and DIR5 (two proteins involved in the lignans biosynthetic pathway) and a lower level of transcripts coding for CHS and CHI, the first two proteins involved in the isoflavonoid biosynthetic pathway. This indicates a shift from the production of isoflavonoids towards the production of lignans. In addition, two transcripts of the proteins involved in the production of raffinose (*DIN10* and *SIP1*) and two transcripts involved in the biosynthesis of trehalose (*TPS* and *TPP*) had their level restored to the control level, although they were downregulated in the roots exposed to cold and metals alone. Finally, the proteins involved in the mitigation and repair of ROS damage during oxidative stress displayed a strong synergetic interaction pattern. These proteins were HSP 90, two elongation factors, and the beta subunit G1 of the 20S proteasome. This could indicate that roots simultaneously exposed to cold and metals rely more on repairing frost-induced oxidative damage when it occurs rather than going through long-term metabolic adjustment to avoid damage, as is observed in the aboveground parts of *Petunia* × *hybrida* [74]. These proteins might be specifically implicated in the tolerance to multiple stresses and could be targeted for further characterisation and targeted breeding. 

## Figures and Tables

**Figure 1 ijms-25-01545-f001:**
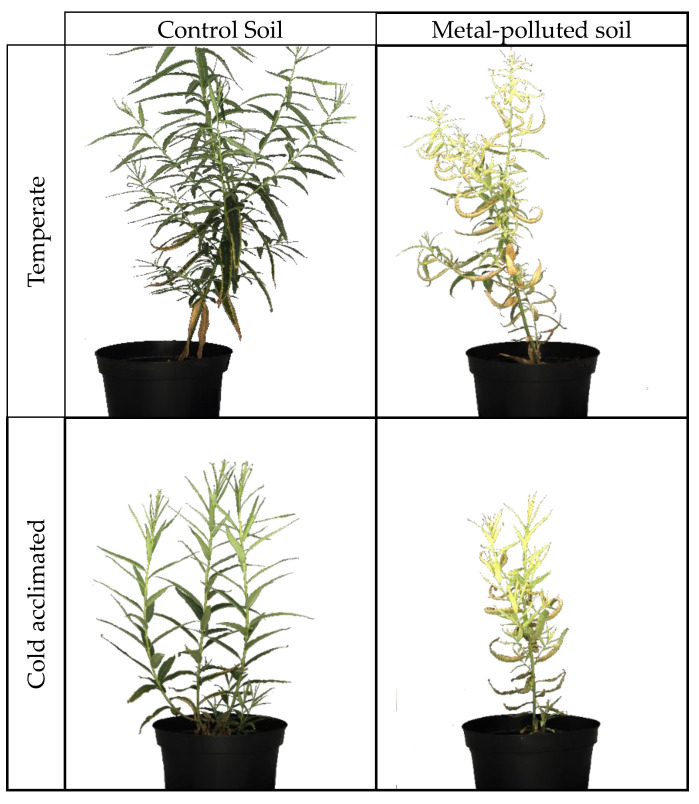
Phenotype of *Salix viminalis* exposed to a polymetallic mixture and/or cold-acclimated. Plants were exposed to metals for two months before being cold-acclimated at 7 °C/5 °C for one month.

**Figure 2 ijms-25-01545-f002:**
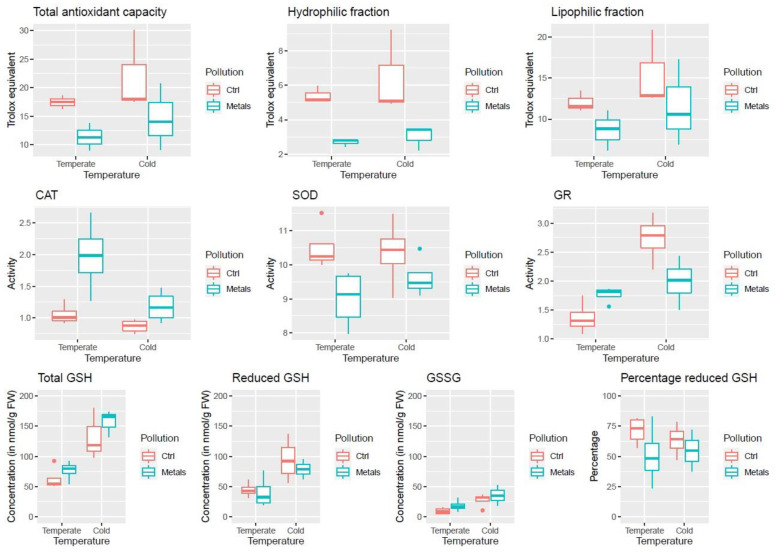
Impact of cold acclimation and heavy metals on root antioxidant system. First row: impact of cold and metals on the antioxidant capacity of the whole extract, hydrophilic fraction, and lipophilic fraction. Second row: impact of cold and metals on the activity of catalase (CAT), superoxide dismutase (SOD), and glutathione reductase (GR). Third row: impact of cold and metals on total glutathione (GSH), reduced glutathione, oxidised glutathione (GSSG), and the fraction of reduced glutathione.

**Figure 3 ijms-25-01545-f003:**
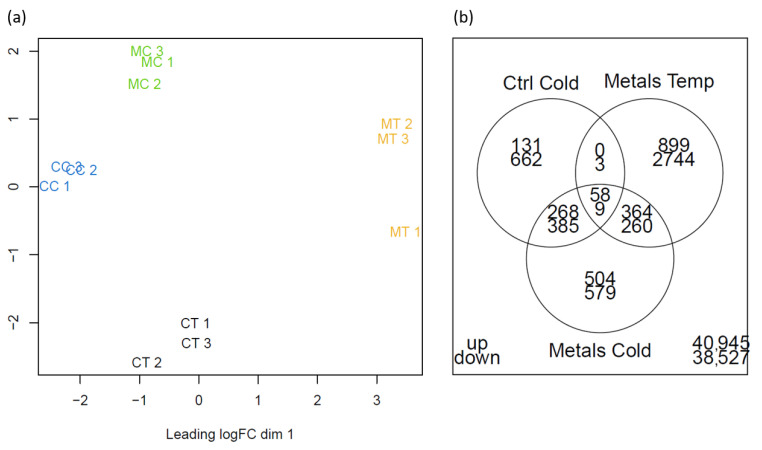
(**a**) Multidimensional scaling plot of the top 500 most significant DEGs. (**b**) Venn diagram of the DEGs in the three conditions investigated. Upper numbers are upregulated genes, lower numbers are downregulated genes.

**Figure 4 ijms-25-01545-f004:**
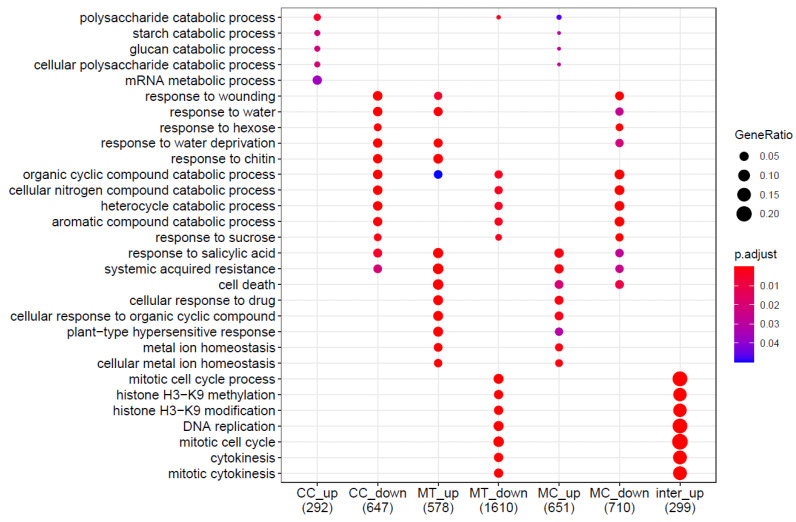
Dot plot of gene ontology (GO) enrichment analysis relative to the CT condition showing the five most over-represented biological processes for each category. Point size is determined by the proportion of all transcripts within a category that were annotated with the GO term (GeneRatio). Colours are based on the adjusted *p*-value of the enrichment analysis.

**Figure 5 ijms-25-01545-f005:**
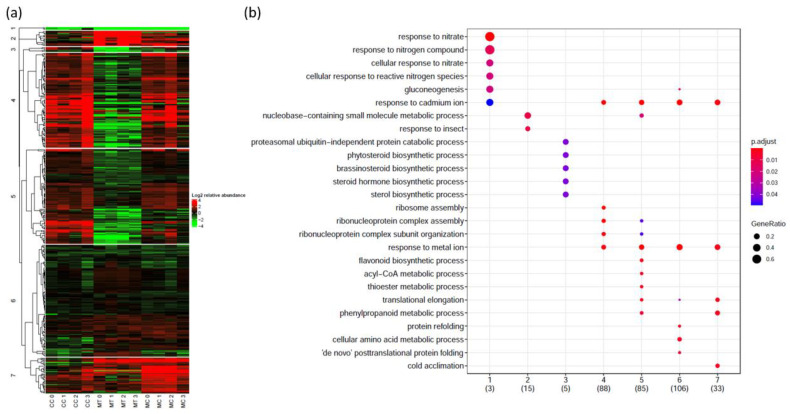
(**a**) Hierarchical clustering of proteins displaying interaction patterns. Clusters were made by cutting at a correlation level of 0.75. For better visualisation, protein abundance was normalised by the protein abundance of CT roots. For each condition, a technical replicate, whose name ends in 0, was made by mixing an equal protein amount of each biological replicate. (**b**) Dot plot of gene ontology (GO) enrichment analysis showing the five most over-represented biological processes for each protein cluster. Point size is determined by the proportion of all proteins within a cluster that were annotated with the GO term (GeneRatio). Colours are based on the adjusted *p*-value of the enrichment analysis.

**Table 1 ijms-25-01545-t001:** Impact of cold acclimation and heavy metals on root antioxidant system.

(1)				
**Factor**		**TAC**	**Hydro.**	**Lipo.**
Temperature		0.175	0.366	0.157
Heavy metals		0.031 *	0.002 *	0.112
Interaction		0.84	0.668	0.914
(2)				
**Factor**		**CAT**	**SOD**	**GR**
Temperature		0.017	0.562	0.003 *
Heavy metals		0.004 *	0.015 *	0.446
Interaction		0.09	0.345	0.004 *
(3)				
**Factor**	**GSH Tot.**	**GSH Red.**	**GSSG**	**Perc. GSH**
Temperature	<0.001 *	0.007 *	0.0127 *	0.746
Heavy metals	0.258	0.529	0.1877	0.132
Interaction	0.723	0.704	0.939	0.58

Two-way ANOVA of the factors explaining antioxidative system variation. Values given are *p*-values. 1. Impact of cold acclimation (CA) and heavy metals (HM) on total antioxidant capacity (TAC), antioxidant capacity of the hydrophilic (Hydro.), and lipophilic (Lipo.) fractions. 2. Impact of CA and HM on catalase (CAT), superoxide dismutase (SOD), and glutathione reductase (GSH) activity. 3. Impact of CA and HM on total glutathione (GSH tot.), reduced glutathione (GSH red.), glutathione disulfide (GSSG) concentrations, and on the percentage of reduced glutathione over total glutathione (Perc. GSH). *: statistically significant. *p* < 0.05

## Data Availability

Raw sequences from the transcriptome experiment have been submitted at the Gene Expression Omnibus as accession GSE218490. The mass spectrometry proteomics data have been deposited at the ProteomeXchange Consortium via the PRIDE [73] partner repository with the dataset identifiers PXD030968 and 10.6019/PXD030968. All other data is either provided in the text or as Appendix A.

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
