# Peer review of "Impact of Heavy Metals on Cold Acclimation of Salix viminalis Roots"

_ijms, 2024, doi:10.3390/ijms25031545_

Round 1
Reviewer 1 Report
Comments and Suggestions for Authors
The manuscript presents lots of data, but not in a clear manner.
The abstract section is not informative enough to reflect the findings of interest. Rewrite the section.
Why do you test two major stressor effects on the plant? Frost and a mixture of heavy metals. One of these independently hurts the plant.
The introduction section never represents a reliable literature review to describe the whole idea of the study. Please explain in detail the damage of heavy metals and their interaction with low temperatures.
The aims of the study have to be presented more and more clearly.
The conclusion section is designed to represent the highlighted results from the present experiment plus the perspective for future studies. Amend accordingly.
Comments on the Quality of English Language
Minor editing of English language required
Author Response
Reviewer 1 :
- The manuscript presents lots of data, but not in a clear manner.
We rewrote parts of the introduction to better present some previous observations and explain the hypothesis and aims underlying this research. In addition, several sections were changed in order to present the results and discuss them in a clearer way.
- The abstract section is not informative enough to reflect the findings of interest. Rewrite the section.
The abstract section was rewritten to better reflect the findings of the research.
- Why do you test two major stressor effects on the plant? Frost and a mixture of heavy metals. One of these independently hurts the plant.
We tested the interaction of two stressors because although the response of plants’exposure to a single stress has been studied extensively, the response of plants to the simultaneous exposure to multiple stressors is unique and cannot be deduced from the response of the plants exposed to each of the stressor separately. The reason why we choose these two stresses is because a few years ago, our team observed that the roots of Salix viminalis grown in a soil polluted by a mixture of heavy metals were more frost hardy than those of plants grown in unpolluted soil. This background explanation has been added to the manuscript.
- The introduction section never represents a reliable literature review to describe the whole idea of the study. Please explain in detail the damage of heavy metals and their interaction with low temperatures.
We could not find any literature going over the interactions between a mild and long-term polymetallic exposure and cold acclimation. The interactions between theses stressors has been unknown for a while (Suzuki et al., 2014). This is an important factor that motivated this present study.
- The aims of the study have to be presented more and more clearly.
We better presented the aim of the study in the introduction.
- The conclusion section is designed to represent the highlighted results from the present experiment plus the perspective for future studies. Amend accordingly.
We modified the conclusion to better highlight the results from this experiment and present some perspectives.
Reviewer 2 Report
Comments and Suggestions for Authors
I propose incorporating hypothesis testing before stating the aim of the paper. This approach would enhance the logical flow of the manuscript, providing a clearer context for the subsequent research objectives.
-
In line 542, I recommend changing the units to mg/kg of soil. Additionally, please specify the form of metals used and how they were applied.
-
In line 571, please directly indicate the tables/figures in the supplementary material. Furthermore, ensure that this material is well-labelled with captions.
-
In the 2.1 section, the title of this section is not properly related to the content. Consider transferring the photos of the plants from the supplementary material. Why some morphometric analyses of plants were not conducted?
-
Include a note on the visualization of the figures, particularly regarding areas where the font size is too small.
-
In line 238 and in many other places, please do not use statment "significantly" or "higher", "smaler" etc. if appropriate statistical tests have not been performed.
-
Perform two-way ANOVA and multiple comparison tests in relation to the data shown in Figure 4, as well as other data presented in the supplementary material. However, based on the view of Figure 4, I have serious doubts about whether all data meet the assumptions of parametric tests.
-
Ensure that the conclusion and discussion are clearly related to the hypotheses set up at the beginning of the paper.
Author Response
Reviewer 2 :
- I propose incorporating hypothesis testing before stating the aim of the paper. This approach would enhance the logical flow of the manuscript, providing a clearer context for the subsequent research objectives.
Thank you for this advice. We added some observations that we made in the past and lead to this study as well as our working hypothesis in the introduction.
- In line 542, I recommend changing the units to mg/kg of soil. Additionally, please specify the form of metals used and how they were applied.
We changed the units and added some information about the form of metals used as well as how they were added.
- In line 571, please directly indicate the tables/figures in the supplementary material. Furthermore, ensure that this material is well-labelled with captions.
This line referred to supplementary written Material and Method that we forgot to upload. It has been added accordingly.
- In the 2.1 section, the title of this section is not properly related to the content. Consider transferring the photos of the plants from the supplementary material. Why some morphometric analyses of plants were not conducted?
We changed the title and transferred the photos of the plants to this section. The main focus of this article is the transcriptomic and proteomic responses of roots exposed to cold and/or metals. As a consequence, we did not record enough morphological data to be able to perform a valid morphometric analysis.
- Include a note on the visualization of the figures, particularly regarding areas where the font size is too small.
We changed some figures to make them bigger and increased their resolution. It should now be easier to read what is written on the figures.
- In line 238 and in many other places, please do not use statment "significantly" or "higher", "smaler" etc. if appropriate statistical tests have not been performed.
We are sorry, these statements date from before we had to reduce the manuscript to fit the guidelines. We put back the statistical tests where they are needed.
- Perform two-way ANOVA and multiple comparison tests in relation to the data shown in Figure 4, as well as other data presented in the supplementary material. However, based on the view of Figure 4, I have serious doubts about whether all data meet the assumptions of parametric tests.
Two-way crossed ANOVA were added for the data shown in Figure 4 (now Figure 5) but also for the Fv/Fm and the REL data. The use of crossed ANOVA allowed to determine if interaction effects were present. Although the data was not normal, ANOVA is known to be quite robust against the violation of the normality assumption. In addition, when significant, most of the p-values were low (< 0.01), protecting us from false positive.
- Ensure that the conclusion and discussion are clearly related to the hypotheses set up at the beginning of the paper.
We rewrote the conclusion and added sections in the discussion in order to better answer the hypothesis that was set up in the introduction.
Round 2
Reviewer 1 Report
Comments and Suggestions for Authors
I believe that the authors have addressed nearly all the comments raised in the evaluation process and the manuscript after some minor writing and punctuation corrections is worthy of acceptance.
Comments on the Quality of English LanguageI believe that the authors have addressed nearly all the comments raised in the evaluation process and the manuscript after some minor writing and punctuation corrections is worthy of acceptance.
Reviewer 2 Report
Comments and Suggestions for Authors
The paper can be published in the present form.